# The Use of Cluster Analysis to Assess the Threats of Poverty or Social Exclusion in EU Countries: The Case of People with Disabilities Compared to People without Disabilities

**Bożena Frączek** 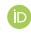

Department of Banking and Financial Markets, University of Economics in Katowice, 40-287 Katowice, Poland; b.fraczek@ue.katowice.pl

**Abstract:** The main research objective was to assess the range of threats related to poverty or social exclusion in EU countries among people with disabilities as compared to people without disabilities. The research used the available poverty determinants, including the percentage of low work intensity, the percentage of low income, the percentage of material deprivation, the poverty risk rate, the percentage of severe disability, living conditions and population income, and the overall risk of household poverty or exclusion, which are available in Eurostat databases. The data used in the research relates to 2018 and was published in 2021. The research used cluster analysis, more specifically one of the agglomeration clustering methods, i.e., Ward's method. Separate cluster analysis using Ward's method was carried out for people with disabilities and for people without disabilities. The analysis identifies two clusters among people with disabilities and two clusters among people without disabilities. In the group of people with disabilities, cluster 2 includes 19 countries with a higher risk of poverty or social exclusion, while cluster 1 includes eight countries with a lower risk of poverty or social exclusion. In turn, in the group of people without disabilities, cluster 2 includes nine countries with a higher risk of poverty or social exclusion, while cluster 1 includes 18 countries with a lower risk of poverty or social exclusion.

**Keywords:** poverty; social exclusion; European Union countries; cluster analysis

## 1. Introduction

The problem of poverty in the world is of great importance and highly relevant nowadays. The trend that had continued for almost 25 years of a steadily declining number of people living in extreme poverty—on less than $1.90 per person per day—was interrupted in 2020 due to the COVID-19 crisis and due to the effects of conflict and climate change—which had already been slowing the reduction in poverty [1]. The global poverty rate (at the poverty line of US $1.90) was 8.6% in 2018, 9.1% in 2017, 10.1% in 2015, and 12.9% in 2012 [2]. Due to the impact of COVID-19 on increasing poverty, in 2020 an estimation was made of the difference in poverty in a world with and without the pandemic. It was estimated (in 2021), that the pandemic had pushed about 97 million additional people into extreme poverty around the globe in 2020. The implications of further estimates are that starting from 2021, global poverty is projected to decline in line with the pre-pandemic trend [3].

The concepts of poverty and social exclusion are very often discussed and explored together, but they do not mean the same. To understand the linkage between these two concepts, definitions for each of them are required. When discussing poverty and social exclusion, the problem is that these concepts and the relation between them are difficult to define.

Poverty does not have one single definition. Some authors even state that it is not possible to define and measure poverty in an "objective" way. This is why many authors and researchers adopt different assumptions and approaches to describing poverty. Generally,

poverty represents deprivation [4]. Other studies present a wider description and consider four approaches to defining poverty: the monetary approach, the capability approach, the participatory approach, and social exclusion [5]. The different approaches enable the types of poverty to be specified and help to measure poverty, while at the same time underlining the relationship between poverty and social exclusion.

The World Bank—due to its mission to end extreme poverty and promote shared prosperity—defines poverty as "the inability to attain a minimal standard of living", and describes poverty through the prism of the financial approach, specifying three levels of poverty (poverty lines): the number of people living in extreme poverty—on less than $1.90 per person per day (most commonly used) as well as two additional levels $3.20 and $5.50 per person per day [2].

Many authors specify the definitional clusters focused on conceptually different meanings of poverty [6]. In addition, poverty as a composite concept embraces a range of meanings and related concepts:

- Poverty as a material concept—this refers to inadequate economic resources and consumption, resulting in a lack of material goods or services such as food, clothing, fuel, or shelter, which make up physical and mental well-being [7,8]. The range of related concepts includes: need, a pattern of deprivation, and limited resources.
- Poverty as economic circumstances (economic terms)—understood through the prism of the standard of living, in which individuals or families are considered poor when their level of living, measured in terms of income or consumption, is below a particular standard. Poverty in this meaning implies the concept of inequality, determining certain degrees or dimensions of inequality. The range of related concepts includes: standard of living, inequality, economic position, and social circumstances (including: social class, dependency, lack of basic security, lack of entitlement, and exclusion).
- Poverty as a moral judgement—this refers to serious deprivation, where people are seen as poor when their material circumstances are deemed to be morally unacceptable. The related concept is unacceptable hardship, which implies the moral imperative that something should be done.

On the one hand, these descriptions of poverty are logically separable and refer to distinct circumstances (poverty as inequality and poverty as a lack of basic security, or poverty as a standard of living and poverty as a dependency). On the other hand, they also overlap—the definitions are linked by family resemblance; the need is closely related to the standard of living, the standard of living is closely related to resources, etc. In addition, the definitions confirm the link between poverty and social exclusion.

In fact, social exclusion is also difficult to define, and the definition depends on the particular country's context. In general, "social exclusion describes a state in which individuals are unable to participate fully in economic, social, political and cultural life, as well as the process leading to and sustaining such a state" [9]. In addition, many scholars expand the UN definition and clarify that social exclusion is a complex and dynamic process that occurs in several dimensions; social, political, economic, and cultural. Scientists usually underline that social exclusion persists across different levels of society, and the notion of society has expanded its meaning to encompass aspects of an ever-globalizing world [10,11]. What is also important is that social exclusion can be seen not just in levels of income, but also as affecting matters such as health, education, access to services, housing, or debt. In some countries (e.g., India, Peru), the fixed castes also result in social exclusion, i.e., inter-caste or inter-religious marriages, people with leprosy, as well as people of indigenous origin, etc., [5].

The relationship between poverty and social exclusion seems to be reciprocal. On the one hand, poverty may result from social exclusion [12]. On the other hand, poverty may create vulnerability to social exclusion [13]. Both poverty and social inclusion may concern the same group of people, as those who are socially excluded are in most cases poor (especially in the multidimensional meaning of poverty). Poverty usually influences the exclusion of people from participation in the normal pattern of social life due to limited

resources [14]. Although a great percentage of the population in a society may be poor, (i.e., suffer from adverse incorporation), this does not necessarily imply they are excluded. However, poverty (one could suggest) influences the exclusion of people from participation in the normal pattern of social life due to limited resources.

Some authors believe that social exclusion is a wider concept than poverty. Social exclusion is seen as a process of progressive social rupture, which—in comparison to poverty—is a more comprehensive and complex conceptualization of social disadvantage/inequality [15]. Social exclusion implies inequality or relative deprivation, whereas poverty need not. Poverty and social exclusion affect particular individuals (households), but also geographical areas and groups of people [16]. A separate, but at the same time very important issue, is the measurement of poverty and social exclusion in showing their scale for selected regions, countries, or groups of people. The diversified structure of various indicators indicates many aspects/dimensions of poverty and social exclusion.

This results in different institutions/organizations, e.g., the World Bank in comparison to the European Union, using quite different ways to measure the level of poverty. Nevertheless, to underline the significance of describing the problem (poverty), it is worth presenting the differences in poverty on a global scale, and subsequently, showing the situation in the EU in the following stages of the paper.

In the geographical dimension, certain regions and countries are affected by the highest level of poverty and social exclusion. The World Bank defines the extreme poor as those living on less than $1.90 a day [2]. The latest estimates presented by the World Bank show that the regional poverty rates calculated for 2019 differ:

- from 1.1% in Europe and Central Asia to 38.4% in Sub-Saharan Africa (at the US $1.90 poverty line);
- from 4.1% in Europe and Central Asia to 64.7% in Sub-Saharan Africa (at the US $3.20 poverty line);
- from 11.5% in Europe and Central Asia to 85% in Sub-Saharan Africa (at the US $5.50 poverty line).

In turn, EU statistics regarding the risk of poverty or social exclusion encompass: (1) low household income—often described as "risk of poverty" and measured as low household income relative to the median of the population; (2) severe material and social deprivation (absolute poverty according to the European minimum standard of living); (3) the lack of or very low intensity of work in the household. One of these components is enough to qualify a person as at risk of poverty or social exclusion. Therefore, the at-risk of poverty or social exclusion rate (AROPE) corresponds to the sum of persons who are either at risk of poverty, severely materially and socially deprived, or living in a household with very low work intensity [17].

In 2020, 21.9% of the population (i.e., 96.5 million people) in the EU was at risk of poverty or social exclusion—in the AROPE meaning. More detailed statistics confirm that among these 96.5 million inhabitants within the EU, 27.6 million were severely materially and socially deprived, and 27.1 million lived in a household with low work intensity, while some 5.9 million (1.3% of the total population) lived in households experiencing simultaneously all three poverty and social exclusion risks (risk of poverty, severely materially and socially deprived, and living in a household with very low work intensity) [18].

In terms of the general population, there are certain groups that are at higher risk of poverty or social exclusion. Among them are usually children [19], women [20], the unemployed [21], and the elderly—with an additional impact of social exclusion on this group being a large number of premature deaths [22]. People with disabilities, meanwhile, deserve special attention in research on poverty and social exclusion.

At this moment, it should be underlined that over one billion people, or 15% of the world's population, experience some form of disability. Among them, there are between 110 million and 190 million people who experience significant disabilities. Persons with disabilities experience adverse socioeconomic outcomes such as less education, poorer health outcomes, lower levels of employment, a lower level of financial inclusion, and

higher poverty rates [23,24]. This, in turn, may result in many cases of a limited range of proper social behaviours (including consumer behaviour) [25].

Although the relationship between disability and poverty has been recognized in the literature since the 1990s, there is still a lack of knowledge about many aspects of this problem [26]. Previous research confirms the strong connection between poverty and disability, mainly in developing countries, and results even underline the need for a separate poverty line for households with members with disabilities in comparison to other households [27]. Meanwhile, in developed countries, people with a disability experience disproportionately high poverty rates [28]. In addition, households with a member with a disability have been recognized as a population at higher risk of poverty in comparison to other households, due to higher levels of multidimensional poverty (measured by the Multidimensional Poverty Index, MPI) [29]. The direct reasons for definitely higher threats of poverty in the group of people with disabilities in comparison to people without a disability are on the one hand higher basic daily living costs (that increase barriers to full participation in society), and on the other lower income due to limited access to the labour market [28].

The EU maintains special statistics on people with disabilities, and similar statistics on people without disabilities. Each member state of the EU is obliged to prepare a report—i.e., a country fiche on disability equality, which includes as part of the document statistics relating to poverty or social exclusion. The latest country fiches on disability equality were prepared by individual countries in European Semester 2020–2021 (published in 2021) and related to 2019. Access to these documents allowed for further analysis and for determining the goals of this research.

The main research objective was therefore to assess the range of threats of poverty or social exclusion in EU countries amongst people with disabilities compared to people without disabilities.

## 2. Materials and Methods

The formulated objective of the research is reflected in the following research questions:

1. Is it possible to specify clusters of EU countries that present different levels of threats of poverty or social exclusion among people with disabilities and people without disabilities, and if so, how many clusters may be specified?
2. Are the clusters resulting from the cluster analysis different or similar among people with disabilities and people without disabilities?

The research is exploratory; therefore, no research hypotheses have been formulated.

Eurostat data was used for the analysis. The data used in the research relates to 2018 and was published in 2021 (the latest data at the moment of conducting the research). The research used cluster analysis, more specifically one of the agglomeration clustering methods, i.e., Ward's method. Separate cluster analysis using Ward's method was carried out for people with disabilities and for people without disabilities.

## 3. Research and Results

The primary data used in the research consists of 27 EU countries and 11 attributes (parameters) for people with a disability and 10 attributes (parameters) for people without a disability (Table 1).

**Table 1.** Primary data used in the research into poverty and social exclusion.

| People with a Disability | People without a Disability |
|---|---|
| People at risk of poverty or social exclusion, by risk | |
| 1. Low work intensity [%]<br>2. Low income [%]<br>3. Materially deprived [%] | 1. Low work intensity [%]<br>2. Low income [%]<br>3. Materially deprived [%] |
| People at risk of poverty or social exclusion, by gender aged 16+ | |
| 4. Some disability [%]<br>5. Severe disability [%]<br>6. Disabled women [%]<br>7. Disabled men [%] | 4. Without a disability [%]<br>5. Nondisabled women [%]<br>6. Nondisabled men [%] |
| Overall risk of household poverty or exclusion by age 16+ | |
| 8. Disabled (16–64) [%]<br>9. Disabled (65+) [%] | 7. Nondisabled (16–64)<br>8. Nondisabled (65+) |
| People at risk of poverty after social transfer | |
| 10. Working age persons (16–64) [%]<br>11. Persons over 65 [%] | 9. Working age persons (16–64) [%]<br>10. Persons over 65 [%] |

The linearity between the variables was checked and the correlated variables were then removed. As a result of correlation analysis of the 11 parameters for people with a disability and the 10 parameters for people without a disability, five were distinguished for each group as not showing statistically significant (at $p < 0.05$) strong correlation relationships (i.e., taking into account the strength of the relationship above 0.75). In the further cluster analysis, the following parameters were used:

For people with disabilities:

LWI [%]—percentage of people with low work intensity,

LI [%]—percentage of people with low income,

MD [%]—percentage of people materially deprived,

SD [%]—percentage of people with a severe disability,

PRWP [%]—percentage of people with a poverty risk rate for working-age persons (16–64) after social transfer

For people without disabilities:

LWI [%]—percentage of people with low work intensity,

LI [%]—percentage of people with low income,

MD [%]—percentage of people materially deprived,

ORPSE [%]—percentage of people in the group with overall risk of household poverty or exclusion (65+)

PR65+ [%]—percentage of people with a poverty risk rate for persons over 65 after social transfer

All the data is available in the Eurostat databases. Separate cluster analysis using Ward's method was carried out for people with disabilities and for people without disabilities. Before the presentation of the research results, it is worth presenting the essence of the parameters used.

Work intensity refers to how much all working-age household members have worked in comparison to their full potential. In detail, work intensity is defined as "the ratio of the total number of months that all working-age household members have worked during the income reference year and the total number of months the same household members theoretically could have worked in the same period". Work intensity is usually shown at three levels, ranging from very high, to medium and very low. Very high work intensity is when the household adults' working time exceeded 85% of the work-time potential.

In turn, very low work intensity means that the household adults had a working time equal to or less than 20% of their total combined work-time potential. The lower the work intensity within a household, the harder (further) it is for household members to attain full employment, and at the same time the higher the probability of being at risk of poverty [30].

The low-income parameter in this research refers to the percentage of working-age persons with an equalized disposable income (after social transfer) below the at-risk-of-poverty threshold, which is set at 60% of the national median equalized disposable income after social transfers. This is known as the at-risk-of-poverty rate and represents the social situation of people. It should be underlined that this indicator measures low income in comparison to other residents in a given country, and not wealth or poverty, which does not necessarily imply a low standard of living [31].

Material deprivation is defined as the enforced inability (and not the choice) to pay unexpected expenses relating to typical daily life or typical family expenses. The material deprivation rate is an EU-SILC indicator that refers to the inability to afford some items considered by most people to be desirable or even necessary to lead an adequate life [32].

The severe disability parameter refers to the proportion of people aged 16 or over with a severe disability who were at risk of poverty or social exclusion

The poverty risk rate for working-age persons (16–64) or for persons over 65 after the social transfer is known as the at-risk-of-poverty rate after social transfers for a given group. This is the percentage of persons in a given population who are at risk of poverty.

The overall risk of household poverty or exclusion (65+) refers to the proportion of people aged 65 or over who were at the overall risk of household poverty or exclusion.

The data used in the cluster analysis was standardized to a mean of 0 and a standard deviation of 1.

The next stage of the research process was hierarchical clustering (Ward's method) in order to group the EU member states. The research answers the two research questions formulated in the method section.

Answer to the first research question: The cluster analysis identified two clusters among people with disabilities and two clusters among people without disabilities. The specified clusters in each group (people with a disability and people without a disability) may be determined as less at risk of poverty or social exclusion, or more at risk of poverty and social exclusion.

For the group of people with disabilities, the first cluster of countries with a lower risk of poverty or social exclusion includes eight countries, while the second cluster of countries with a higher risk of poverty or social exclusion includes 19 countries (Figure 1). Whereas for the group of people without disabilities, the first cluster of countries with a lower risk of poverty or social exclusion includes 18 countries (including all eight countries from the first cluster for people with a disability), and the second cluster of countries with a higher risk of poverty or social exclusion includes nine countries (Figure 2).

The answer to the second research question: In each of the two clusters specified for people with a disability and for people without a disability there are both differences as well as similarities. The first differences were mentioned above and relate to the different number of countries in each cluster, wherein in the case of people with disabilities there are more countries in the second cluster, which is characterized by a higher risk of poverty and social exclusion. It is worth noting that the levels of particular parameters describing poverty or social exclusion in the groups of people with and without disabilities are also quite different: they are much higher in the case of people with a disability. In turn, the differences and similarities both relate to the countries that belong to a given cluster (less or more at risk of poverty or social exclusion) among people with and without disabilities (Tables 2 and 3).

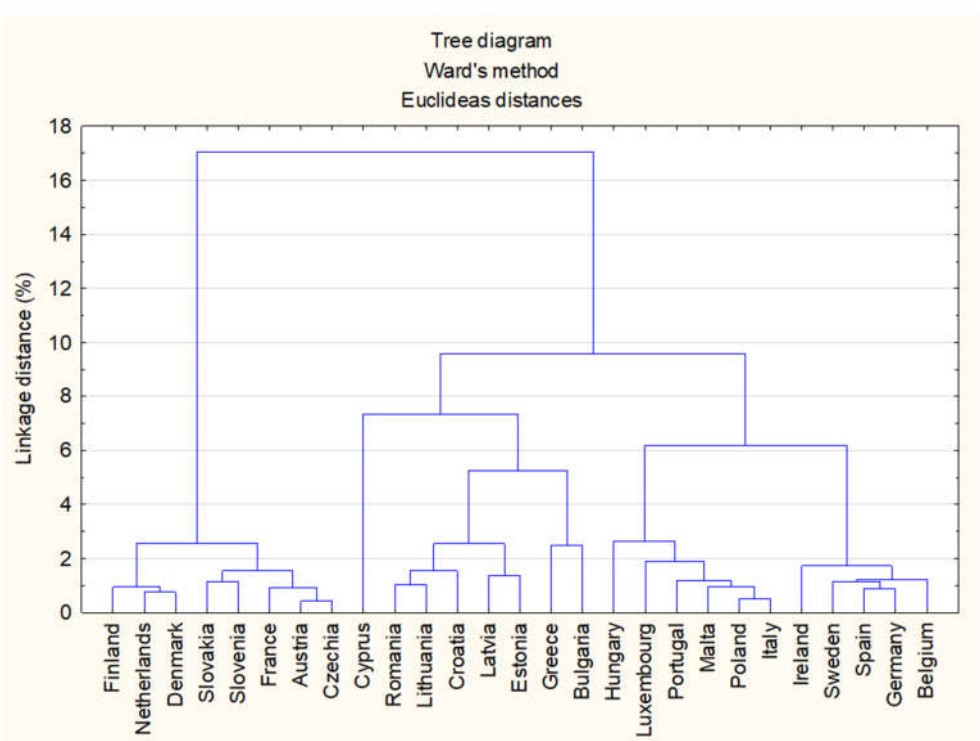

**Figure 1.** Hierarchical cluster analysis for people with disabilities. Linkage method—Ward's method. Euclidean distance of all elements.

**Table 2.** Lowest, highest, and average parameters describing poverty or social inclusion in the two specified clusters among people with a disability in EU countries.

| | Low Work Intensity | Low Income | Materially Deprived | Severe Disability | Poverty Risk Rate for Working Age Persons (16–64) after Social Transfer |
|---|---|---|---|---|---|
| First cluster (eight countries)—lower level of parameters (least at risk of poverty or social exclusion) | | | | | |
| **Finland, Netherlands, Denmark, Slovakia, Slovenia, France, Austria, Czechia** | | | | | |
| Min | 10.40 | 14.50 | 5.60 | 24.40 | 14.50 |
| Max | 20.80 | 18.40 | 10.50 | 39.20 | 18.40 |
| Weighted average | 17.58 | 16.99 | 8.28 | 29.08 | 16.99 |
| Second cluster (19 countries)—higher level of parameters (most at risk of poverty or social exclusion) | | | | | |
| Cyprus, **Romania, Lithuania**, Croatia, **Latvia, Estonia, Greece, Bulgaria**, Hungary, Luxembourg, Portugal, **Malta**, Poland, **Italy**, Ireland, Sweden, **Spain**, Germany, Belgium | | | | | |
| Min | 15.10 | 20.80 | 3.00 | 30.40 | 22.00 |
| Max | 35.20 | 31.70 | 31.60 | 58.90 | 31.70 |
| Weighted average | 24.76 | 26.67 | 13.42 | 38.10 | 26.68 |

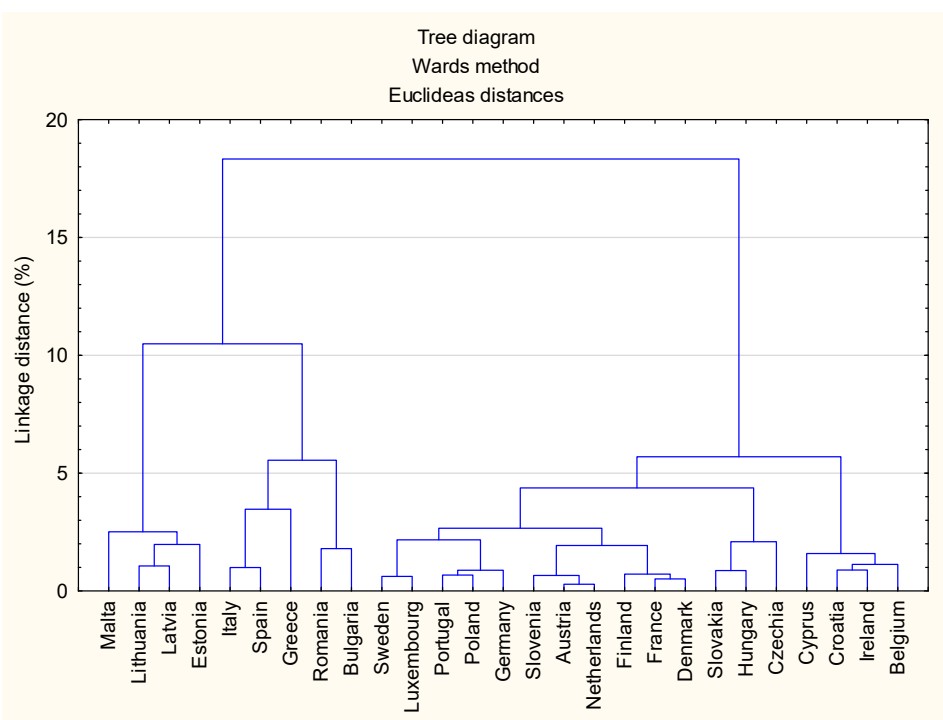

**Figure 2.** Hierarchical cluster analysis for people without disabilities. Linkage method—Ward's method. Euclidean distance of all elements.

**Table 3.** Lowest, highest, and average parameters describing poverty or social inclusion in the two specified clusters among people without a disability in EU countries.

| | Low Work Intensity | Low Income | Materially Deprived | Overall Risk of Household Poverty or Exclusion (65+) | Poverty Risk Rate for Persons over 65 after Social Transfer |
|---|---|---|---|---|---|
| First cluster (18 countries)—lower level of parameters (least at risk of poverty or social exclusion) | | | | | |
| Sweden, Luxembourg, Portugal, Poland, Germany, **Slovenia, Austria, Netherlands, Finland, France, Denmark, Slovakia**, Hungary, **Czechia**, Cyprus, Croatia, Ireland, Belgium | | | | | |
| Min | 2.90 | 7.20 | 1.00 | 5.70 | 3.90 |
| Max | 9.30 | 16.00 | 9.00 | 22.50 | 18.70 |
| Weighted average | 5.58 | 15.99 | 3.23 | 12.69 | 11.78 |
| Second cluster (9 countries)—higher level of parameters (most at risk of poverty or social exclusion) | | | | | |
| **Malta, Lithuania, Latvia, Estonia, Italy, Spain, Greece, Romania, Bulgaria** | | | | | |
| Min | 2.90 | 12.80 | 2.40 | 14.40 | 10.00 |
| Max | 14.30 | 21.10 | 16.60 | 40.70 | 39.10 |
| Weighted average | 9.53 | 20.19 | 8.76 | 18.97 | 14.96 |

specified clusters amo.

The analysis of the parameters presented in Tables 2 and 3 shows that every parameter describing a given aspect of poverty or social exclusion (used in the cluster analysis) in every presented aspect (min, max, weighted average) is higher for the countries that belong to the second cluster in comparison to the countries that belong to the first cluster. This means that the risk of poverty or social exclusion is higher for countries that belong to

the second cluster for both groups (people with and without a disability). In addition, the levels of lowest and highest risk of poverty or social exclusion for people with disabilities are generally much higher than for people without disabilities (Figure 3).

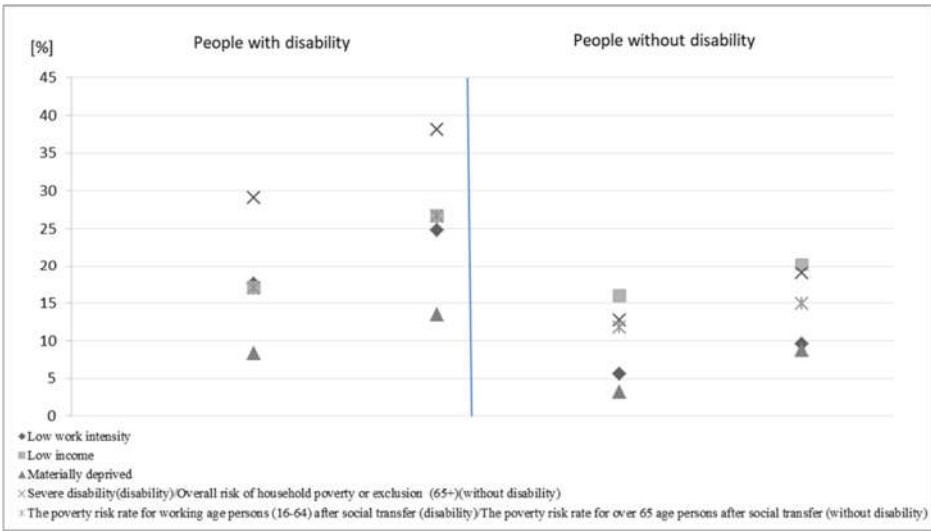

**Figure 3.** Levels of lowest and highest risk of poverty and social exclusion parameters for people with and without a disability in EU countries.

## 4. Discussion

Although the EU region is generally viewed as having a relatively low risk of poverty or social exclusion in comparison to other regions in the world, and despite European Union policies directed toward combatting poverty and social exclusion due to income inequalities, there is still a large variation in this respect in individual EU countries. These differences concern both people with disabilities and people without disabilities, with the first group being significantly more at risk. The cluster analysis conducted using the hierarchical Ward's method allows for more detailed observations, which may be used in individual countries to pursue appropriate policies aimed at preventing poverty and social exclusion.

Specifying the clusters of countries with a lower or higher risk of poverty or social exclusion among people with and without a disability may be a starting point for a very wide multidimensional discussion. First of all, it is worth paying attention to which countries are the most and the least at risk of poverty or social exclusion.

Taking into account the sum of the values (percentage values) of the parameters included in the research, i.e., those describing the level of risk of poverty or social exclusion, the values from the highest to the lowest were achieved by the following countries [33–59]:

Among people with a disability:

Cluster 2 (more at risk of poverty or social exclusion) includes 19 countries: Bulgaria, Ireland, Greece, Lithuania, Latvia, Croatia, Germany, Romania, Spain, Belgium, Cyprus, Estonia, Sweden, Hungary, Italy, Poland, Luxembourg, Portugal, Malta.

Cluster 1 (less at risk of poverty or social exclusion) includes eight countries: Netherlands, Denmark, Finland, Czechia, Slovenia, France, Austria, Slovakia.

Among people without a disability:

Cluster 2 (more at risk of poverty or social exclusion) includes nine countries: Bulgaria, Latvia, Lithuania, Estonia, Romania, Greece, Malta, Italy, Spain.

Cluster 1 (less at risk of poverty or social exclusion) includes 18 countries: Croatia, Ireland, Cyprus, Germany, Portugal, Belgium, Poland, Sweden, Slovenia, Luxembourg, Netherlands, Hungary, Finland, Austria, Denmark, Czechia, France, Slovakia.

It turns out that among people with disabilities, the most unfavourable situation in terms of the risk of poverty or social exclusion is found in Bulgaria, Ireland, and Greece

(the highest total value of the analysed parameters in the group of countries characterised by a greater risk of poverty or social exclusion). These countries were also included in the cluster of countries with a higher risk of poverty or social exclusion among people without disabilities. The highest values of the parameters describing poverty and social exclusion among people without a disability were recorded in Bulgaria, Latvia, and Lithuania.

At the other extreme, i.e., countries with a lower risk of poverty or social exclusion, are Slovakia, Austria, and France—among people with disabilities, and Slovakia, France, and Czechia—among people without disabilities. These countries achieved the lowest total value of the parameters describing the risk of poverty or social exclusion.

The analysis shows that Bulgaria turned out to be the country with the highest risk of poverty or social exclusion, while Slovakia turned out to be the country with the lowest risk (both among people with disabilities and people without disabilities).

The unfavourable situation in the field of poverty or social exclusion and related areas in Bulgaria, as well as the relatively good situation in Slovakia, together with the relevant recommendations of the UN CRPD, are confirmed by documents prepared by these countries, i.e., European Semester 2020–2021 country fiche on disability equality [35,56]. It is worth noting that both countries belong to the group of Central and Eastern European countries, i.e., countries with a relatively lower level of development compared to highly developed EU countries.

The use of two clusters means that there are similarities among countries belonging to a given cluster, and differences between countries from different clusters. The similarities may be the result of the European social model, in turn, the differences may be due to individual EU member states developing their own systems, which affects the diversity in the level of poverty or social exclusion in particular countries.

A more detailed but still general analysis at this stage of the research shows that the clusters of countries specified in the research do not correspond to the groups of countries on the basis of the development level criterion (highly developed and developing countries) and related criteria, e.g., taking into account the level of social benefits.

Preliminary analysis shows that the emerging clusters relating to the diversity of the risk of poverty or social exclusion more closely correspond to the level of employment and the disability gap [33–59]. However, these are areas for the next stages of further, more detailed research.

Nevertheless, it is undisputed that people with a disability are definitely more exposed to poverty or social exclusion than people without a disability (in every EU member state).

## 5. Conclusions and Implications

The standardized way data is presented on the level of poverty and social exclusion in EU countries, as well as the conclusions drawn from this data and recommendations, enabled the conducting of this research. The effect is the selection of clusters of countries where the degree of association/relations among particular countries in terms of risk of poverty or social exclusion is high if they belong to the same group, and low if they belong to different groups.

The main objective of this paper was to assess the diversity of risk of poverty or social exclusion in EU countries among people with disabilities compared to people without disabilities. To achieve this objective, hierarchical clustering by Ward's method was used.

Despite the common values represented in the European social model, particular EU member states have developed their own individual systems, which affects the diversity in the level of poverty or social exclusion in particular countries.

Additional analysis confirms that the two specified clusters do not correspond to the groups of countries determined on the basis of the development level criterion and level of social benefits, and more closely correspond to the level of employment and the disability gap.

The research also confirms that people with a disability are clearly more exposed to poverty or social exclusion than people without a disability (in every EU member state):

Among people with a disability, a higher risk in terms of poverty or social exclusion was identified in 19 countries: Bulgaria, Ireland, Greece, Lithuania, Latvia, Croatia, Germany, Romania, Spain, Belgium, Cyprus, Estonia, Sweden, Hungary, Italy, Poland, Luxembourg, Portugal, and Malta, while a lower risk of poverty or social exclusion was identified in eight countries: Netherlands, Denmark, Finland, Czechia, Slovenia, France, Austria, Slovakia.

In turn, among people without a disability, a higher risk of poverty or social exclusion was identified nine countries: Bulgaria, Latvia, Lithuania, Estonia, Romania, Greece, Malta, Italy, and Spain, and a lower risk of poverty or social exclusion was identified in 18 countries: Croatia, Ireland, Cyprus, Germany, Portugal, Belgium, Poland, Sweden, Slovenia, Luxembourg, Netherlands, Hungary, Finland, Austria, Denmark, Czechia, France, Slovakia.

What is interesting in both groups (people with a disability and people without a disability) is that the highest risk in terms of poverty or social exclusion was found in Bulgaria, and the lowest risk in terms of poverty or social exclusion was in Slovakia. In addition, the research confirms that a disability still translates into a greater risk of poverty or social exclusion in comparison to those without a disability. In every EU Member State, people with a disability were exposed to a higher risk of poverty or social exclusion. As is well known, people with limitations on their activity rely heavily on social transfers, and the preliminary analysis showed that the countries with a higher risk of poverty or social exclusion are not the same as the countries with the highest level of development and level of social protection benefits.

The research should be continued further. The next step should be an in-depth study/analysis of the relationship between the risk of poverty or social exclusion and the level of development of a country, the level of social benefits, the level of employment, the employment disability gap, economic activity, etc.

As the next step in the research, the comparative analysis should also be carried out into solutions applied via social policies and their effectiveness in individual countries.

**Funding:** This research received no external funding.

**Institutional Review Board Statement:** Not applicable.

**Informed Consent Statement:** Not applicable.

**Data Availability Statement:** The data used in the research relates to 2018 and was published in 2021 in documents presented in References section under items [30–56].

**Conflicts of Interest:** The authors declare no conflict of interest.

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
