# Peer review of "The Use of Cluster Analysis to Assess the Threats of Poverty or Social Exclusion in EU Countries: The Case of People with Disabilities Compared to People without Disabilities"

_sustainability, doi:10.3390/su142114223_

Round 1

Reviewer 1 Report

In general terms I believe this is a very good and informative and coherent study on the issue of poverty and social exclusion in relation to people with and without disabilities. The author begins with a clear definition of poverty and social exclusion, phenomena that seem to complicated to define. My only concerns are the following: 

The following sentences in pp.2-3 need citation: 

1) 'The relationship between poverty and social exclusion seems to be reciprocal. On the one hand, poverty may result from social exclusion, while on the other hand poverty may create vulnerability to social exclusion.' Although it might seem reasonable to suggest that poverty and social exclusion are reciprocal, for a scientific study it would be important such claims to be backed by evidence (perhaps the author could make references to the findings of another study). In addition, I would rephrase the second sentence: 'On the one hand, poverty may result from social exclusion. On the other, poverty may create vulnerability to social exclusion (citation)'. It would be better to avoid repetition of words and phrases: on the one hand, on the other hand... Better write: on the one hand... on the other... 

2) 'Poverty usually influences the exclusion of people from participation in the normal pattern of social life due to limited resources.' Certainly I agree with this statement. But (as above) references are required here. I would certainly rephrase the whole next two sentences. Here there is a good (but improved) example: 'While a great percentage of a population in a society are poor, (i.e. suffer from adverse incorporation), this does not necessarily imply they are excluded. However, poverty (one could suggest) influences the exclusion of people from participation in the normal pattern of social life due to limited resources.' I have, thus, improved the flow. At the same time, I have removed words like the 'majority' or 'usually'. The latter seems too vague and imprecise. The former requires referencing; claims about 'the majority of a population' or a social group (affected by the X or Y policies et al) are too strict to be made without evidence to support them. In the event there is no data to support such a claim, the author could come with a logical argument/opinion. Words such as 'the majority' are too certain to be used without scientific references. 

3) Likewise: 'Social exclusion implies inequality or relative deprivation, whereas poverty need not. Poverty and social exclusion affect particular individuals (households), but also geographical areas and groups of people.' This sentence requires citation. 

Furthermore, I can see that the author brings to the discussion findings and stats from the World Bank, show that regional poverty rates calculated for 2019 differ...' I stand a bit confused here. Why do we need evidence from central Asia and sub-Saharan Africa? Is not Europe the central focus? I can see in section 4 (the general discussion of the findings) the reason the author makes references to these findings. It is a comparative strategy (if I understand correctly). I suggest, however, to add a few sentences before mentioning statistics from Asia and Africa in order to alert the reader. In order words, the study requires better signposting as the reader becomes confused at some point with all these different (albeit valuable) data presented. 

Finally, concerning the methods, for a thesis that relies on data I am uncomfortable to comment. Were this set up as a study in political science, engaging with theoretical issues, I would be less uncomfortable. 

Overall: I believe this is a very good piece of work. I would like to see it published. It requires, however, some minor/moderate revisions in order to further shine. 

Author Response

Dear Reviewer,

Thank you very much for all your valuable comments. All your suggestions have been taken into account. I would like also to thank you for appreciation of my efforts in research on assessing the threats of poverty or social exclusion in EU countries in the case of people with disabilities compared to people without disabilities. 

The new literature was added according to the suggested citation.

The suggested changes were made in the text.

All changes in the manuscript are marked in green to highlight every change.

Author

Reviewer 2 Report

Thank you for allowing me to review the study. In my opinion, this is a very important topic and should therefore definitely be published. However, I have a few comments. 

1)    The introduction is too general and deals too long with the topic of the definition. What is the state of research on the topic of people with disabilities and poverty? This part is too short.

2)    From my point of view, the article has too little theoretical foundation. Even though it may be an exploratory study, there must be some idea or explanation of how these differences come about. Finally, there is also a simple explanation: what is the position of people with disabilities in the social system or in society (opportunities for education, etc.).

3)    The conclusion needs to focus more on explanations for the clusters. Why is this so? What could be the reasons? What does it mean?

From my point of view, the article shows important aspects. However, it lacks a theoretical basis and explanations. The article should be improved in this respect.  However, the topic is certainly important and a publication of a revised version would be desirable.

Author Response

Dear Reviewer,

Thank you very much for all your valuable comments. All your suggestions have been taken into account. I would like also to thank you for appreciation of my efforts in research on assessing the threats of poverty or social exclusion in EU countries in the case of people with disabilities compared to people without disabilities. 

The aspect of disability and poverty was developed together with an explanation of the reason for the differences between the level of poverty of people with a disability and people without a disability. The additional material is supported by the appropriate literature.

The conclusion has been expanded to include more explanations for the clusters. 

All changes in the manuscript are marked in green to highlight every change.

Author

Reviewer 3 Report

The article is very interesting because it assesses the various threats associated with poverty or social exclusion in EU countries among people with disabilities compared to people without disabilities.

The determinant variables used are also complex, namely existing poverty, including the percentage of low work intensity in the city, the percentage of low income, the percentage of material shortages, the level of poverty risk, the percentage of severe disability, living conditions and income of the population, and the overall risk of poverty or house exclusion. stairs, which are available in the Eurostat database

Abstract, it is quite clear, explains the background, objectives and benefits of the research clearly, the method used in the form of the Ward Method is also clear and describes the results of the research clearly.

Introduction, especially in the background, the author has very clearly outlined the problems, objectives and benefits of this paper, however, the author needs to reaffirm the significance of the research or the urgency of this research.

The method has also been described using a separate cluster analysis using the Ward's method carried out for people with disabilities and for people with disabilities, it's very clear and good.

Results and Discussion are sufficiently strong to describe the poverty-related or social threats among people with disabilities compared to people without disabilities. However, it needs to be strengthened and reaffirmed novelty from the results of discussions and discussions.

From grammar, grammar, spelling are very good, only need to be improved, the same vocabulary appears, maybe need to find another vocabulary or paraphrase.

References are complete with updated sources, both books, articles and journals. Just need to check again whether it is in accordance with the template and connect with the citation in the article.

Author Response

Dear Reviewer,

Thank you very much for all your valuable comments. All your suggestions have been taken into account. I would like also to thank you for appreciation of my efforts in research on assessing the threats of poverty or social exclusion in EU countries in the case of people with disabilities compared to people without disabilities. 

The conclusion was strengthened and novelty was taken into account from the results of the discussion section.

The previous as well as current (updated) references have been checked in accordance with the template and linked to the citations in the article.

All changes in the manuscript are marked in green to highlight every change.

Author

Round 2

Reviewer 2 Report

In summary, I would say that the article was well reworked in some aspects. However, my Review 1 seems to have disappeared. 

Therefore, I have to state again that a more theoretical determination and argument would be very important.

1)    The introduction is too general and deals too long with the topic of the definition. What is the state of research on the topic of people with disabilities and poverty? This part is too short. Even in the revision, this part is too superficial.

2)    From my point of view, the article has too little theoretical foundation. Even though it may be an exploratory study, there must be some idea or explanation of how these differences come about. Finally, there is also a simple explanation: what is the position of people with disabilities in the social system or in society (opportunities for education, etc.).

3)    The conclusion needs to focus more on explanations for the clusters. Why is this so? What could be the reasons? What does it mean?

The idea of the work is good - without question. Ultimately, however, the work is descriptive. At least to some extent, the work must also investigate the "why".

Author Response

Dear Reviewer, 

Thank you for all your reviews. I added additional short text regarding people with disability in Introduction section. I would like to explain, that in my concept this general level of description is enough for this article.
I would like also explain that simultaneously for few months I have been working on next article (this will be continuation of this article) on suggested by reviewer education (in the group of persons with and without disabilities) in EU.

Thank you 
Author